# Microglia Depletion-Induced Remodeling of Extracellular Matrix and Excitatory Synapses in the Hippocampus of Adult Mice

**DOI:** 10.3390/cells10081862

**Published:** 2021-07-22

**Authors:** Luisa Strackeljan, Ewa Baczynska, Carla Cangalaya, David Baidoe-Ansah, Jakub Wlodarczyk, Rahul Kaushik, Alexander Dityatev

**Affiliations:** 1Molecular Neuroplasticity, German Center for Neurodegenerative Diseases (DZNE), 39120 Magdeburg, Germany; Luisa.Strackeljan@dzne.de (L.S.); Carla-Marcia.Cangalaya-Lira@dzne.de (C.C.); David.Baidoe-Ansah@dzne.de (D.B.-A.); 2Nencki Institute of Experimental Biology, Polish Academy of Sciences, Pasteura 3, 02-093 Warsaw, Poland; e.baczynska@nencki.edu.pl (E.B.); j.wlodarczyk@nencki.edu.pl (J.W.); 3Institut für Biochemie und Zellbiologie, Medical Faculty, Otto-von-Guericke-University, 39120 Magdeburg, Germany; 4ESF International Graduate School on Analysis, Imaging and Modelling of Neuronal and Inflammatory Processes, 39120 Magdeburg, Germany; 5Center for Behavioral Brain Sciences (CBBS), 39106 Magdeburg, Germany; 6Medical Faculty, Otto-von-Guericke University, 39120 Magdeburg, Germany

**Keywords:** extracellular matrix, synapses, parvalbumin, brevican, microglia, perineuronal nets, perisynaptic ECM

## Abstract

The extracellular matrix (ECM) plays a key role in synaptogenesis and the regulation of synaptic functions in the central nervous system. Recent studies revealed that in addition to dopaminergic and serotoninergic neuromodulatory systems, microglia also contribute to the regulation of ECM remodeling. In the present work, we investigated the physiological role of microglia in the remodeling of perineuronal nets (PNNs), predominantly associated with parvalbumin-immunopositive (PV+) interneurons, and the perisynaptic ECM around pyramidal neurons in the hippocampus. Adult mice were treated with PLX3397 (pexidartinib), as the inhibitor of colony-stimulating factor 1 receptor (CSF1-R), to deplete microglia. Then, confocal analysis of the ECM and synapses was performed. Although the elimination of microglia did not alter the overall number or intensity of PNNs in the CA1 region of the hippocampus, it decreased the size of PNN holes and elevated the expression of the surrounding ECM. In the neuropil area in the CA1 *str. radiatum*, the depletion of microglia increased the expression of perisynaptic ECM proteoglycan brevican, which was accompanied by the elevated expression of presynaptic marker vGluT1 and the increased density of dendritic spines. Thus, microglia regulate the homeostasis of pre- and postsynaptic excitatory terminals and the surrounding perisynaptic ECM as well as the fine structure of PNNs enveloping perisomatic—predominantly GABAergic—synapses.

## 1. Introduction

The neural extracellular matrix (ECM) constitutes the non-cellular component of brain tissue, contributing up to 20% of the organ’s total volume [1]. The composition of the brain ECM varies, but in general, the core constituents are the glycan hyaluronic acid and chondroitin sulfate proteoglycans (CSPGs) of the lectican family such as aggrecan, brevican (Bcan), neurocan, and versican (Vcan). These CSPGs are linked to a hyaluronic acid backbone and these links are stabilized via hyaluronan and proteoglycan link proteins (HAPLN1-4). Moreover, lecticans are interconnected through oligomeric glycoproteins of the tenascin family [2,3]. Based on localization and function, the neural ECM can be categorized as either perineuronal nets (PNNs) or the perisynaptic ECM. Whereas PNNs are a specialized form of condensed ECM predominantly enwrapping the soma and proximal dendrites of parvalbumin (PV) expressing fast-spiking GABAergic interneurons in the hippocampus, the perisynaptic ECM associated with synapses on pyramidal neurons is a more diffuse form of neural ECM, which is broadly present in the neuropil area [2,4]. The neural ECM modulates multiple Ca^2+^ and K^+^ channels and plays a bidirectional role in either the support or suppression of synaptic plasticity [5,6,7]. The appearance of PNNs coincides with the closure of the so-called critical period, a phase of enhanced plasticity in the developing brain. PNNs are assumed to stabilize neuronal circuits during development and support the function of established synaptic connections [8,9]. On the other hand, the ECM can act as a structural barrier inhibiting the formation of new synaptic connections after central nervous system (CNS) injury [10,11]. It has been shown that in mouse models of dementia, enzymatic degradation of the ECM with chondroitinase ABC (chABC) can improve synaptic transmission, plasticity, object recognition memory, and contextual fear memory [12,13]. The precise molecular mechanisms are still elusive and currently under investigation.

Microglia serve as resident macrophages in the CNS and contribute up to 10% of the total cell number in the CNS, being densely populated in the hippocampal area [14,15]. Microglia are crucial during brain development and later on for brain homeostasis. They perform a lot of essential functions under physiological conditions, including synaptic pruning during development, the remodeling of mature synaptic connections, the phagocytosis of cellular debris, and the secretion of signaling molecules and proteases in the extracellular space [16,17,18,19].

Microglia perform surveillance of their surroundings with their long and versatile processes monitoring synapses and eliminate some synaptic components. This non-cell-autonomous mechanism of synaptic remodeling is still under investigation but suggests a key role for microglia in diverse forms of synaptic plasticity [20]. For their survival, microglia highly depend on the colony-stimulating factor 1 (CSF1) signaling pathway [21]. The absence of CSF-1 receptors in *Csf1r**^−/−^* mice results in the loss of microglia at an embryonic stage and death before adulthood [22]. Taking advantage of the fact that CSF-1R expressed in microglia is crucial for the survival of these cells, several studies have shown that treatment with CSF-1R inhibitors depletes the brain of microglia and hence is a powerful tool to study microglial function in a variety of biological processes, including the modulation of the neural ECM [21,23]. Interestingly, transcriptomics data collected using the highly specific CSF-1R inhibitor PLX5562 revealed alterations in the mRNA levels of ECM molecules after long-term microglia depletion, including the upregulation of neurally synthesized ECM proteoglycan *Bcan* [24]. This indicates an involvement of microglia in the homeostasis of the neural ECM.

Here, we intended to study the link between physiological microglia functions and the state of the brain ECM in the adult hippocampus. We know that microglia regulate these structures and via these mechanisms shape brain physiology and pathology. However, because PNNs and the perisynaptic ECM embed different types of neurons and synaptic contacts, it is important to study both of these ECM forms separately and in more detail to understand how microglia depletion affects these ECM structures and eventually influences synaptic functions in multiple ways. To that end, we treated 4-month-old transgenic Cx3cr1Cre/Tomato x GFPM Thy-1 mice, with fluorescently labeled microglia and a subset of principal neurons, with the CSF1-R inhibitor PLX3397 (pexidartinib) or vehicle and performed confocal imaging to investigate changes in the ECM. We observe that microglia depletion results in a significant increase in the mean intensity of vGluT1 puncta as well as perisynaptic Bcan in the CA1 region of the hippocampus. Moreover, several alterations are found in the PNN fine structure after the depletion of microglia, highlighting an interesting interplay between microglia, the ECM, and the organization of different subtypes of synapses.

## 2. Materials and Methods

### 2.1. Animals

All animal experiments were conducted in accordance with ethical animal research standards defined by the Directive 2010/63/EU, the German law and the recommendations of the Ethical Committee on Animal Health and Care of the State of Saxony-Anhalt, Germany (license number: 42502-2-1346). Eight male Cx3cr1Cre/Tomato x GFPM Thy-1 mice, which expressed red fluorescent protein (tdTomato) in all microglial cells and enhanced green fluorescent protein (EGFP) in a subset of cortical pyramidal neurons, were used in the study as described previously [20]. Animals were housed individually under standard conditions, including *ad libitum* access to food and water and a 12 h light/12 h dark cycle.

### 2.2. PLX 3397 Treatment

The animals were treated according to the previously published protocol [20]. To induce the recombination that labels microglia with the red fluorescent protein tdTomato, all mice were injected intraperitoneally with tamoxifen at a concentration of 100 mg/kg of body weight at P30 for 5 consecutive days [25]. The PLX3397 treatment started when mice were 4 months of age. To induce microglia depletion, mice were fed with control Nutella or Nutella containing PLX3397 (pexidartinib; MedChemExpress, Monmouth Junction, NJ, USA) at 1000 mg/kg of food for one month as described previously [20,26]. The daily dose was between 1–1.5 mg of PLX3397 in 1–1.5 g of Nutella. 

The sample size was chosen according to the power calculations for the *t*-test with power = 0.8, *p* = 0.05 and the reported values of standard deviation (SD) for spine density (12% in [27]) and major parameters characterizing the shape, size and intensity of perineuronal net units (<12%) [28,29]. We estimated that with n = 4 per group we would be able to detect >30% effects of PLX on these parameters. Additionally, our previous Monte Carlo simulations showed that differences in spine shape can be reliably detected with n = 4 [30]. Moreover, our previous analysis in the retrosplenial cortex with n = 4 mice per group revealed multiple PLX effects on the parameters characterizing the interaction between microglia and synapses {20]. Considering that the standard deviation of perisynaptic brevican expression measurements is about 20% [31], having 4 mice per group would allow us to detect >50% differences in perisynaptic ECM with power = 0.8 and *p* = 0.05.

### 2.3. Tissue Preparation

For tissue preparation, animals were injected with ketamine (90 mg/kg body weight) and xylazine (18 mg/kg body weight) in 0.9% NaCl solution and then transcardially perfused with PBS, followed by 4% paraformaldehyde (PFA). Brains were dissected and fixed in 4% PFA in phosphate-buffered saline (PBS) at 4 °C overnight. The next day, the tissue was transferred to 30% sucrose in PBS overnight before being frozen in methylbutane at -80 °C. Coronal brain sections of 40-µm thickness were prepared using a cryostat at −20 °C and stored in a cryoprotectant solution (ethylene glycol-based: 30% ethylene glycol, 30% glycerol, 10% 0.2 M sodium phosphate buffer pH 7.4, in dH2O) at 4 °C.

### 2.4. Immunohistochemistry

All sections were washed 3 times with 120 mM phosphate buffer (PB), pH = 7.2, for 10 min at room temperature, followed by permeabilization with 0.5% Triton X-100 (Sigma-Aldrich Inc. T9284, St. Louis, MO, USA) in phosphate buffer for 10 min, followed by application of blocking solution—PB supplemented with 0.1% Triton X-100 and 5% normal goat serum (Gibco 16210-064, Amarillo, TX, USA/Thermo Fisher Scientific 16210-064, Waltham, MA, USA)—for 1 hr at RT. Afterward, sections were either incubated overnight at 37 °C or for 2 days at 4 °C depending on which primary reagents were used (described below). The sections were then washed 3 times and incubated with a cocktail of fluorophore-conjugated secondary antibodies. Labeled sections were washed again with PBS and mounted on glass slides using Fluoromount medium (Sigma-Aldrich F4680, St. Louis, MO, USA).

### 2.5. Antibodies and Lectin

The following primary reagents were used: chicken anti-PV (1:500, Synaptic Systems, Göttingen, Germany, 195006), biotinylated *Wisteria floribunda agglutinin* (WFA, 1:1000, Vector laboratories B-1355, Burlingame, CA, USA), rabbit anti-Bcan [32,33] (1:1000), guinea pig anti-vGluT1 (1:1500, Synaptic Systems, 135304), mouse anti-Pcan (1:500, The Developmental Studies Hybridoma Bank, Iowa City, IA, USA, 3F8-c), mouse anti-Vcan (1:500, The Developmental Studies Hybridoma Bank, 12C5-c), rabbit anti-vGAT (1:500, Synaptic Systems), mouse anti-Iba1 (1:1000, Millipore, Boston, MA, USA, MABN92) and rabbit anti-Iba1 (1:1000, Wako, Richmond, VA, USA, 019-19741).

As secondary antibodies, we used Alexa Fluor 405 goat anti-chicken IgG (1:500, Abcam, Cambridge, UK, ab175675), Alexa Fluor 647 goat anti-chicken IgG (1:1000, Invitrogen, Waltham, MA, USA, A-21449), streptavidin Alexa Fluor 405 conjugated (1:1000, Invitrogen, S32351), Alexa Fluor 647 goat anti-rabbit IgG (1:1000, Invitrogen, S32351), Alexa Fluor 405 donkey anti-guinea pig IgG (1:500, Sigma Aldrich, St. Louis, MO, USA, SAB4600230), Alexa Fluor 647 goat anti-mouse IgG (1:1000, Invitrogen, A21236), Alexa Fluor 405 donkey anti-rabbit IgG (1:500, Abcam, ab175651), Alexa Fluor 488 goat anti-rabbit IgG (1:200, Abcam, ab150077), and Alexa Fluor 546 goat anti-guinea pig IgG (1:200, Invitrogen, A11074).

### 2.6. Confocal Microscopy and Image Processing

Images were acquired using a Zeiss confocal microscope (LSM 700). For overall PNN (WFA+) count in the hippocampus, images were acquired at the 10× objective (N.A. = 0.3). The 63×/oil objective (N.A. = 1.4) was used for higher resolution synaptic and ECM imaging. The acquiring conditions were maintained throughout all the imaging sessions. Brain regions were determined according to the Allen Mouse Brain Atlas. All images were then further processed and analyzed using the open software ImageJ (Fiji) [34].

### 2.7. ECM Analysis Around Parvalbumin-Expressing Cells

WFA was used to label PNNs. For the counting of PNNs, one image of representative hippocampal CA1 area per animal was acquired at 10× objective. The numbers of WFA+ and PV+ cells in the CA1 region of the hippocampus were counted manually using the Cell Counter plugin in ImageJ. To examine potential changes in PNN intensity, structure, or composition, we obtained single-cell images at a 63× magnification and as a z-stack of 7 planes at the interval of 0.3 µm and measured the ECM mean intensity around PV cells using WFA labeling or immunostaining against Vcan.

PNNs were imaged and analyzed using a semi-automatic APNU Fiji script to measure the properties of PNN units (holes and surrounding ECM barriers) described elsewhere [28]. The program prompts the user to choose a centrally located pixel within a hole and then the program provides an automatic outline of the PNN hole and surrounding [28]. The output table contains information about key features of the PNN units, such as the size and shape of the holes in 2D and 3D, as well as the intensity of the ECM surrounding the holes.

### 2.8. Analysis of Individual Spine Morphology

The quantitative morphometric analysis of dendritic spines in the CA1 *stratum radiatum* was performed as described previously [35] using custom-written SpineMagick software [36]. The analysis was performed on secondary dendrites using length, head width, area, and a scale-free parameter (length to head width ratio). The spine length was determined by measuring the curvilinear length along the dendritic spine virtual skeleton, which was obtained by fitting the curve (i.e., the fourth-degree polynomial). The fitting procedure was performed by looking for a curve along which integrated fluorescence was at a maximum. The head width of a dendritic spine was defined as the diameter at its widest point, and the bottom part of the spine (1/3 of the spine length adjacent to the dendrite) was excluded. Dendritic segments from 4 animals (10 cells/animal) were morphologically analyzed, resulting in 1764 and 2079 spines per group. The dendritic length of each morphologically analyzed segment was calculated using SpineMagick software. To avoid the effect of distinctly bent dendrites on the total dendritic length, we measured curvilinear length along the analyzed dendritic segment, which was obtained by fitting of n-order polynomial. The spine number was counted manually by a trained neurobiologist in the ImageJ software and 3D Viewer plugin in ImageJ software for density analysis in 2D and 3D, respectively. The statistical analysis was performed using animal-nested *t*-test.

### 2.9. Analysis of vGluT1 Puncta and Perisynaptic Brevican

To study the synaptic changes after microglia ablation in the adult mouse hippocampus and correlate these changes to the respective perisynaptic ECM, we developed a novel semi-automatic open-source Fiji-script that we named Analysis of PeriSynaptic Matrix (APSM_v1.1). The program can be applied to confocal z stacks and prompts the user to choose the sharply focused plane and allow the user-defined confocal layers to be used for quantification. The program detects the localized high-intensity pixels (synaptic boutons) in a noisy background in brain sections. It further allows filtering out puncta that the user may not consider a synaptic bouton based on the intensity, size, and circularity of the structure. This tool provided us the automatic segmentation of synaptic boutons more similar to manual segmentation as compared to other available thresholding methods in Fiji, and allowed us to reliably quantify the individual synaptic puncta and the corresponding perisynaptic ECM. For our study, we used the vesicular glutamate transporter 1 (VGluT1) as a marker for excitatory presynaptic terminals and Bcan as the perisynaptic constituent of ECM. vGluT1 is present in the membranes of synaptic vesicles and functions as a glutamate transporter to maintain synaptic efficacy and has been shown to correlate with the presynaptic vesicle release probability. Bcan is a major proteoglycan of the brain ECM that has been implicated in several brain functions. We quantified the expression of these proteins in the *str. radiatum* and *str. oriens* of the CA1 region of the hippocampus.

Briefly, after the manual selection of the z-plane with the sharpest vGluT1 signal, the program performs a maximum intensity projection of 3 z-planes (the selected central plane + the one above and one below), subtracts the background, and applies a Gaussian filter. Next, it implements Find Maxima algorithm (prominence = 2, exclusion of maxima on image edges) to locate local maxima that are the brightest vGluT1 positive areas of synaptic boutons and collect the x-y coordinates of these pixels (central pixels). Furthermore, the program iterates over all the identified central x-y coordinates and finds the pixels around them to identify the neighboring pixels that belong to the same presynaptic bouton based on user-defined criteria of percentage of the intensity of central pixels within a particular radius around this pixel. For this study, all neighboring pixels within a 1µm radius of the central pixel that are at least 75% as bright as the central pixel are considered to be part of the same bouton. Furthermore, it applies the area (>0.05 μm^2^) threshold to filter out non-well-defined structures. Finally, it measures the properties of the remaining synaptic puncta and also measures the mean intensity signals within perisynaptic bands surrounding the synaptic puncta (width = 0.2 μm in this study) in all channels of the original image.

### 2.10. Statistical Analysis

All bar graphs are expressed as the mean ± standard error of the mean (SEM). Statistical analysis was performed using GraphPad Prism 8.0 (GraphPad Software Inc., La Jolla, CA, USA) and XLSTAT software (2020.5.1; Addinsoft Inc, New York, NY, USA). Gaussian distributions of the data points were verified using the Kolmogorov–Smirnov, Shapiro–Wilk, or D’Agostino tests. Pairwise comparisons were performed using a two-tailed Student’s t-test for Gaussian distributions unless otherwise stated; in case the data did not follow a Gaussian distribution, the Mann–Whitney test was employed. Comparison of brevican expression between groups was performed using the one-tailed t-test for the alternative hypothesis that the intensity is bigger in the PLX-treated than in control mice because the previous study revealed an increase in total brevican expression after PLX treatment [24]. A principal factor analysis was performed to detect the most informative parameters of PNN units. Spearman coefficients of correlation were computed to detect the correlation between measured parameters, the Kolmogorov–Smirnov test was used for comparison of cumulative frequency distribution functions, and two-way analysis of variance (ANOVA, animal-nested) was used to investigate the interdependencies and differences between variables. Statistical significance was set at *p* < 0.05.

## 3. Results

### 3.1. PLX3397 Administration Depletes Microglia in the Brain

Cx3Cr1-Thy1 mice were treated with the CSF1-R inhibitor PLX3397 for 28 consecutive days. To quantify the effect of PLX3397 treatment on microglia depletion, we counted tdTomato- and Iba1-positive microglial cells in the hippocampal CA1 area in the control and treatment group. One week of drug administration resulted in the depletion of more than 90% of microglia in the retrosplenial cortex [20], whereas after 4 weeks of treatment virtually all microglia were depleted from the brain, including the hippocampus (Figure 1A,B).

### 3.2. Microglia Depletion Does Not Alter the Number and Intensity of WFA-Labeled Cells

A minor fraction of the brain’s CSPGs assembles in PNNs that can be visualized by biotinylated *Wisteria floribunda* agglutin (WFA), a commonly used marker for PNN maturity that labels the N-acetylgalactosamine moiety of chondroitin sulfate glycosaminoglycans. Therefore, alterations in PNNs can be measured by changes in the intensity of WFA. Additionally, we measured the changes in the number or intensity of PV+ cells. Thus, to investigate the effect of microglia depletion on PNNs, we counted the number of WFA+ as well as PV+ cells in the CA1 region of the hippocampus. We could not detect any difference in the PNN number between the PLX-treated (73 ± 5.5 PNNs/mm^2^) and the control group (75 ± 13.5 PNNs/mm^2^) (Figure 2A,D). Next, we measured the intensity of WFA around PV+ cells to investigate the prominence of the ECM surrounding these cells. We did not observe any significant changes in the mean cell intensity of either PV (1.0 ± 0.08 vs. 1.3 ± 0.2; *t*-test, *p* = 0.13) or WFA (1.0 ± 0.2 vs. 0.8 ± 0.06; *t*-test, *p* = 0.52) signals in the CA1 region (Figure 2B,D). Additionally, we measured the intensity of the CSPG Vcan that is enriched in PNNs in the hippocampus. There was also no difference between the intensity of Vcan+ PNNs in control and PLX groups (1.0 ± 0.3 vs. 1.009 ± 0.3; *t*-test, *p* = 0.49) (Figure 2C,D).

### 3.3. Microglia Depletion Changes the Properties of PNN Units

Although the depletion of microglia did not result in significant changes in either the number or the total intensity of PNNs, we hypothesized that there might be fine structural changes that occurred at the level of individual PNN units, i.e., ECM-negative holes surrounded by ECM barriers. These individual PNN units encapsulate the synaptic contacts and can be viewed as PNN compartmentalization units that might determine the properties of embedded synapses and heterosynaptic communication. Hence, we performed a single PNN unit analysis as described previously in detail [28]. In total, we measured the properties of 2126 single PNN units from 27 cells (Figure 3A). The analysis was restricted to PNNs surrounding PV+ interneurons.

The principal factor analysis revealed that the size and intensity of PNN units are the major factors determining the variability of PNN units (Figure 3C). The scatter plot of all PNN units in the space of the first two principal factors, which corresponded to the WFA intensity and size, showed a big overlap between units from control and PLX-treated animals but also reveals a fraction of PNN units with a low intensity of ECM that disappears after PLX treatment (Figure 3B). The analysis of cumulative distribution functions of two major unit parameters, such as the hole area and mean intensity in 3D, revealed that PNN holes are smaller, while peaks of ECM expression around them are more prominent after PLX treatment (Figure 3D) (*p* < 0.0001 and *p* < 0.0001, respectively, Kolmogorov–Smirnov test). Additionally, the analysis of the coefficient of variation revealed that PNN units after PLX treatment are more uniform per animal regarding the size (Figure 3E) (*t*-test, *p* = 0.001).

### 3.4. PLX Treatment Leads to an Increase in Presynaptic vGluT1 and Perisynaptic ECM

Microglia have been implicated in interactions with excitatory synapses in young adult brains and found to affect the formation and elimination of dendritic spines and presynaptic filopodia [20]. Hence, we analyzed the effects of microglia depletion on the dendritic spine density, the expression of presynaptic excitatory marker vGluT1 and the associated ECM. As a marker of the ECM, we choose Bcan, which is abundantly present in the CA1 region and has been shown to support LTP in the excitatory CA3-CA1 synapses in the CA1 *str. radiatum* [37]. The analysis of mean intensities of Bcan and vGluT1 (per ROI) revealed an increase in the vGluT1 intensity in CA1 *str. radiatum* (1 ± 0.08 vs. 1.32 ± 0.09; *t*-test, *p* = 0.047), whereas the intensity in CA1 *str. oriens* remained unchanged (1 ± 0.14 vs. 1.22 ± 0.1; *t*-test, *p* = 0.26). The Bcan mean intensity per ROI was not significantly altered after microglia depletion (Figure 4A,B).

Bcan is one of the most abundant CSPGs in the adult brain, present in a soluble form and accumulating at PNNs, the axon initial segments and perisynaptic ECM. The local changes in the levels of Bcan surrounding individual synapses might modulate their properties. To measure the levels of perisynaptic Bcan at excitatory synapses and correlate them to the expression of vGluT1, we developed a Fiji script. After conventional thresholding methods failed to discriminate properly between single synapses in the tissue (Figure 4D), we developed an original approach, as described in Materials and Methods, to automatically detect excitatory presynaptic vGluT1+ puncta and the surrounding Bcan+ bands (Figure 4C,E). PLX-treated animals showed a significantly increased mean intensity of vGluT1 per presynaptic punctum in the CA1 *str. radiatum* (1 ± 0.12 vs. 1.39 ± 0.10; *t*-test, *p* = 0.043), whereas the number (45.2 ± 1.8 /100 μm^2^ vs. 47.3 ± 0.4 /100 μm^2^; *t*-test, *p* = 0.33) and size (0.4 ± 0.03 μm^2^ vs. 0.5 ± 0.01 μm^2^; *t*-test, *p* = 0.18) of vGluT1 puncta remained unchanged (Figure 4F). Analysis in *str. oriens* did not yield any differences in the mean intensity (1 ± 0.15 vs. 1.19 ± 0.1; *t*-test, *p* = 0.33), density (87 ± 4.3 /100 μm^2^ vs. 89 ± 1 /100 μm^2^; *t*-test, *p* = 0.68) or size (0.39 ± 0.01 μm^2^ vs. 0.41 ± 0.01 μm^2^; *t*-test, *p* = 0.29) of vGluT1 puncta (Figure 4F). The intensity of Bcan around presynaptic vGluT1+ puncta was increased in *str. radiatum* after microglia depletion (1 ± 0.17 vs. 1.43 ± 0.1; *t*-test, *p* = 0.036) but not in *str. oriens* (1 ± 0.14 vs. 1.12 ± 0.05.1; *t*-test, *p* = 0.46) (Figure 4D,E). This is not surprising, as there are fundamental differences between *str. oriens* as compared to *str. radiatum*. For instance, excitatory synapses on apical and basal dendrites in *str. radiatum* and *oriens*, respectively, differ significantly in extracellular and intracellular cascades that control synaptic gain: only synapses in *str. radiatum* show a dependency on matrix metalloproteinase-9, which is known to be expressed in microglia [38].

Analysis of the relationship between the expression of vGluT1 and brevican in individual synapses in *str. radiatum* revealed that synaptic enrichment of vGluT1 coincided with a stronger perisynaptic Bcan signal (Figure 4G). Using a model with the Bcan intensity as a linear function of treatment, vGluT1 intensity, and interaction between them, significant effects of treatment (*p* < 0.0001) and interaction between treatment and vGluT1 intensity (*p* < 0.0001) were detected. The model provided an excellent fit to experimental data with R^2^ = 0.79. The slope of the regression line approximating the Bcan–vGluT1 relationship was smaller in the PLX-treated group than in the control (0.906 + 0.015 vs. 1.274 + 0.018, respectively; Figure 4G). After microglia depletion, there was an increase in the intensity of perisynaptic Bcan at synapses with a low vGluT1 signal. These observations are in line with our analysis of PNN units where a fraction of low-intensity ECM disappeared after PLX treatment.

### 3.5. Microglia Depletion Increases Spine Density but Did Not Change Spine Morphology

To investigate structural changes in dendritic spines in the CA1 *str. radiatum*, we performed a spine density analysis as well as a morphometric analysis of individual spines (Figure 5). Microglia depletion led to an increase in spine density (95.1 ± 2.7 vs. 121.7 ± 2.4 spines /100 μm; nested *t*-test, *p* = 0.0023) (Figure 5B,C). The total numbers of 1764 spines in the control group and 2079 spines in the PLX group were analyzed to compare morphological features. There were no changes in the length, head width, or area of spines after microglia depletion (Figure 5D–G).

## 4. Discussion

In the present study, we investigated the role of microglia in the adult mouse brain. Our study supports the hypothesis that microglia play an important role in the homeostasis of synapses and the ECM in the adult mouse brain. Although the elimination of microglia does not alter the level of PV expression and the overall number of PV+ cells, most of which are associated with PNNs in the hippocampus, the fine structural analysis of individual PNN units revealed that the depletion of microglia decreased the size of PNN holes, increased the intensity of the surrounding ECM, and reduced the heterogeneity of PNN units. As the ECM in PNNs is predominantly perisynaptic for GABAergic contacts, and PNN components tenascin-R and neurocan have been shown to modulate the number of such contacts [39,40], the changes in PNN unit parameters are expected to affect inhibitory synapses. Moreover, we show that microglia modulated excitatory synaptic homeostasis in the hippocampus by affecting both pre- and postsynaptic terminals and their surrounding ECM in the neuropil area.

### 4.1. Microglia and Perisynaptic ECM

Previous reports revealed a strong increase in PNN-related WFA intensity following microglia depletion in the subiculum and visual cortex of C57Bl/6J mice [23,24]. Here, we focused on the CA1 region of the hippocampus of healthy 4–5-month-old mice as this region is highly involved in learning and memory.

A gene expression analysis in the hippocampus revealed that after 6 months of treatment with the specific CSF1R-inhibitor PLX5622, the mRNA levels of *Bcan* were upregulated in the hippocampus of wild-type and the thalamus of 5xFAD mice, while the expression of aggrecan remained unchanged [24]. At a protein level, using immunohistochemistry, simple measurements of mean intensities in ROIs located in the neuropil did not allow us to detect PLX effects on the Bcan expression. However, more advanced image analysis revealed the accumulation of perisynaptic Bcan.

While previous studies focused on the effects microglia have on PNN components, little is known about the interaction of microglia and the perisynaptic ECM on excitatory neurons, which makes the major fraction of the brain’s ECM [41]. Bcan is present in most PNNs but not obligatory for their formation, even though PNNs appear less prominent in mice lacking this lectican as compared to wild types [37]. In contrast to other CSPG family members, Bcan expression is highly localized in the neuropil area enclosing presynaptic terminals, peaks later during development, and is also stable in the mature brain [42]. Whilst its function is still not entirely understood, there is evidence that Bcan plays a key role in regulating synaptic plasticity since *Bcan*-deficient mice show an impairment of LTP maintenance in the CA1 region in hippocampal slices [37]. Additionally, Bcan regulates the excitatory input to PV+ interneurons and modulates their electrophysiological properties by interaction with potassium channels and AMPA receptors [7]. Considering Bcan function and location, we developed a tool that allowed us to study the intensity of Bcan more precisely around single presynaptic boutons. Generally, we show an increase in the intensity of both presynaptic vGluT1 and perisynaptic Bcan in the CA1 *str. radiatum*. At the single synapse level, our analysis shows a positive correlation, i.e., high-intensity vGluT1 puncta are surrounded by high-intensity perisynaptic ECM. After microglia depletion, fractions of low-intensity perisynaptic ECM and PNN units vanish, indicating that they may represent “remainings” after microglia “attacks”. Interestingly, these changes in ECM and presynaptic organization, but not in the density of presynaptic terminals, were accompanied by an increase in the spine density, suggesting that microglia depletion may increase the proportion of so-called multiple spine synapses, i.e., synaptic configurations in which one bouton is in contact with more than one spine [43]. Interestingly, microglial activation under inflammatory conditions may lead to opposite changes, namely to the instability and loss of dendritic spines [44]. Increased spine density after microglia depletion suggests that these cells play a crucial role in the homeostatic maintenance of dendritic spine densities during adulthood [45]. During development, microglia eliminate synapses via the complement pathway that mainly involves secreted C1q, C3 and the C3Ra microglial receptor [16]. It is plausible that this pathway may also mediate microglia’s role in the maintenance of dendritic spines in adult animals.

As a cautionary note, it is important to mention that the depletion of microglia has been shown to induce no changes in GFAP+ cell numbers or morphology, but it increased the expression of GFAP, suggesting an increased level of astrocytic activation [21]. Activated (possibly by dying microglia) astrocytes may secrete more CSPGs, which may then accumulate perisynaptically in line with the observed increase in perisynaptic Bcan after microglia depletion. Moreover, one can not exclude other indirect effects of PLX on perisynaptic ECM and excitatory synapses, e.g., via the regulation of peripheral immunity, oligodendrocyte progenitors and neurons, some of which express CSF1R receptors [46,47,48]. However, the most parsimonious explanation is that microglia directly control the perisynaptic ECM and further live imaging studies are warranted to directly investigate the interaction between microglia, synapses and the perisynaptic ECM.

In the developing dLGN, microglia are known to be directly involved in synaptic pruning in an activity-dependent manner [16]. Additionally, in the adult brain, they participate in synapse turnover via phagocytosis [18,49]. Even though there are several candidates under investigation, including different cytokines and complement factors, the question of how exactly microglia select and engulf these presynaptic terminals remains unanswered [20,49,50]. Additionally, it has been shown that microglia processes can very specifically target the extracellular space in the neuropil area and introduce local extracellular remodeling [51]. If this interaction is direct or indirect and how changes in microglia might affect the brain ECM under physiological conditions have been addressed in several recent studies. One work has explored the cytokine IL-33 as a potential signaling molecule to drive ECM engulfment by microglia, as a loss of IL-33 resulted in ECM accumulation around synapses and dendritic spines. Additionally, IL-33 cKO mice had significantly more total Bcan and less cleaved Bcan in the dentate gyrus [19]. These results are supported by our finding of elevated brevican in the CA1 *str. radiatum* of PLX-treated mice. Furthermore, we could establish a link between ECM and vGluT1 intensities. Possibly Bcan functions as a gatekeeper shielding the synapses and therefore hindering phagocytic activity by microglia. Since weaker synapses also display weaker Bcan coating, they might be more easily engaged in remodeling. Bcan turnover in the adult brain is characterized by degradation by matrix metalloproteases released from neurons and glial cells [52,53]. This indicates an involvement of microglia in the homeostasis of the neural ECM, potentially via both phagocytosis and secretion of matrix metalloproteinases and proteases of the ADAMTS family that control the turnover of the ECM [31]. Additionally, microglia may affect the ECM and synapses via other secreted factors such as cytokines [54].

### 4.2. Microglia Depletion Decreases Heterogeneity in PNNs

Contrary to another study, we did not detect an overall increase in the numbers of PNNs after microglia depletion [23], which may be related to the modest sample size used in the present study. However, at the fine structure level, we observed robust alterations in the organization of PNN units. This indicates that microglia depletion in the healthy brain has rather delicate effects on the perisomatic ECM in the hippocampal region of the brain. After microglia depletion, single PNN units appear to become more uniform with regard to size, suggesting a role for microglia in disrupting the homeostasis of the PNN structure.

### 4.3. Concluding Remarks

In conclusion, we have identified an interaction between microglia and the perisynaptic ECM in the hippocampus of healthy young mice, as evidenced by the increase in perisynaptic Bcan after microglia depletion. Our findings also indicate that this interaction is essential for the maintenance of the perisomatic ECM specifically with the fine structure of PNNs. Taken together, these findings highlight an interaction between microglia and ECM homeostasis. While we and others have shown that the depletion of microglia can influence different forms of ECM in different ways, the complex underlying mechanisms remain elusive. In light of our findings, further experiments that precisely study this interplay and how it is regulated under healthy and pathological conditions are needed. This is of special interest as both the ECM and microglia are affected in disease and under neuroinflammatory conditions such as aging.

## Figures and Tables

**Figure 1 cells-10-01862-f001:**
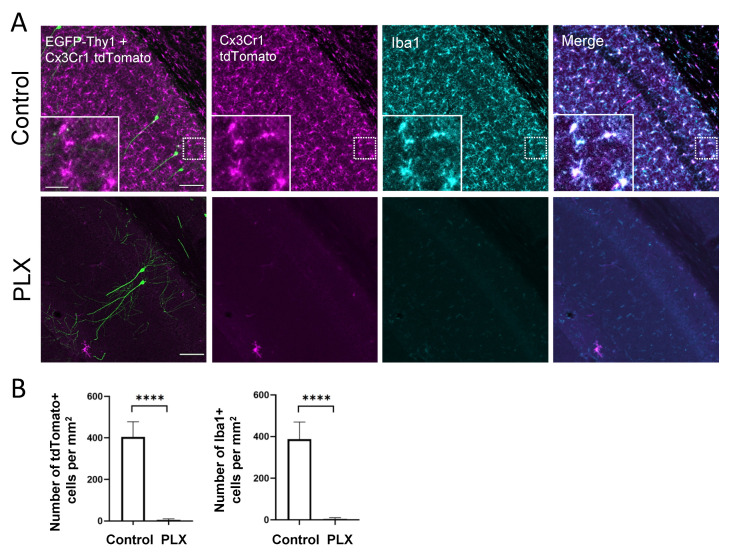
PLX treatment for 28 days leads to microglia depletion. (**A**) The confocal images show perfect colocalization of tamoxifen-induced tdTomato (magenta) signal in Cx3Cr1-expressing cells and immunohistochemical staining for microglial marker Iba1 (cyan) in the CA1 region of the hippocampus. Insets illustrate the typical microglial morphology of labeled cells. EGFP labels a small subset of pyramidal neurons (green), which were used for dendritic spine analysis. (**B**) The bar graphs display the efficacy of the oral PLX3397 administration. After 4 weeks of treatment, virtually all microglia are depleted from the hippocampus. **** *p* <0.0001, *t*-test. Scale bars are 100 µm for main images and 25 µm for inserts.

**Figure 2 cells-10-01862-f002:**
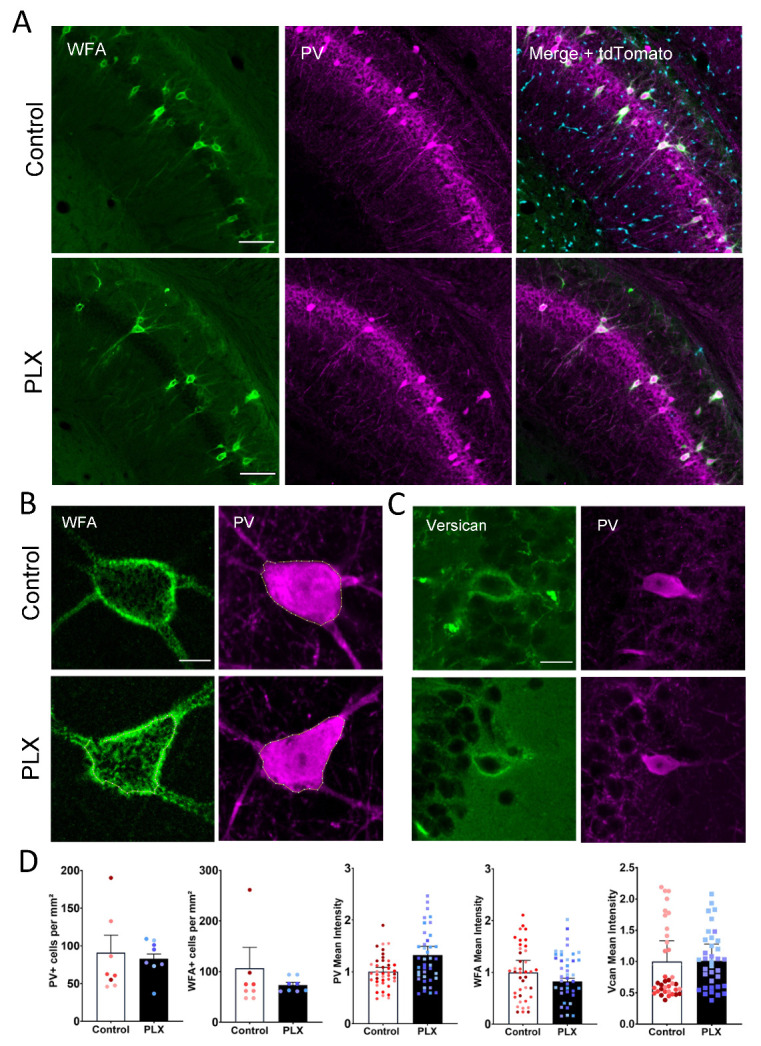
No detectable changes in the density of perineuronal nets after depletion of microglia. (**A**) Representative low (objective ×10) and high resolution (**B**, objective ×63) confocal images showing tdTomato (cyan), WFA+ (green) and PV+ (magenta) cells in the hippocampus of 4-month-old Cx3Cr1-Cre mice. (**C**) Representative 63x images of control and PLX-treated hippocampi stained for PV and versican. (**D**) Bars show mean + SEM values per animal with superimposed individual values per cell. PLX treatment does not affect the number of PV+ or PNN+ cells. No changes in PV, WFA or Vcan mean intensities in PV+ cells could be detected (data from 4 mice in each group). Scale bars, 100 µm (**A**), 10 µm (**B**) and 20 µm (**C**).

**Figure 3 cells-10-01862-f003:**
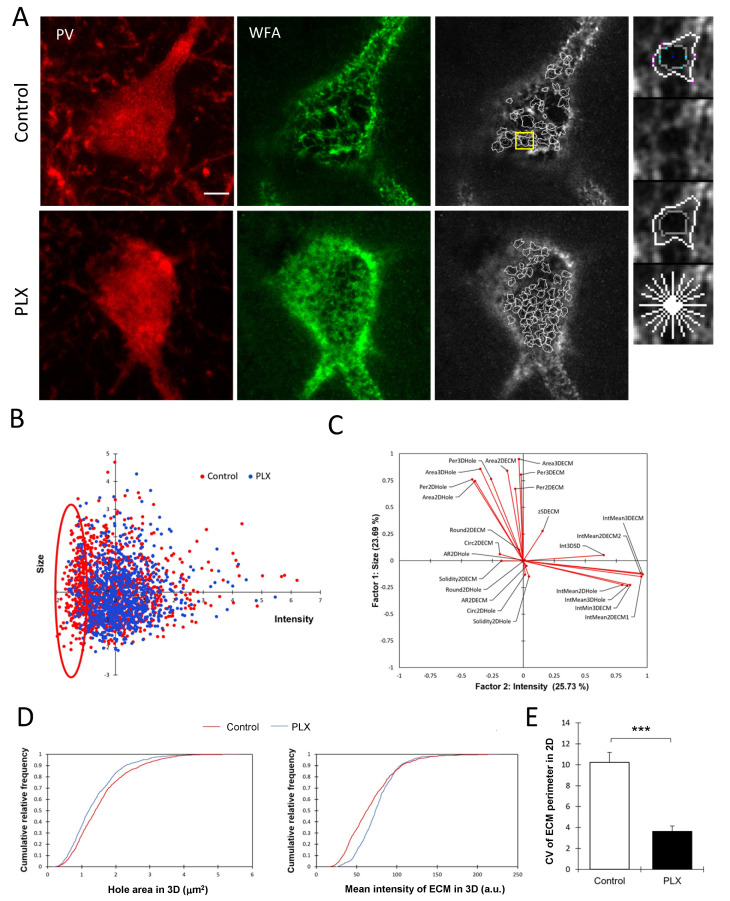
Alterations of PNN fine structure after depletion of microglia. (**A**) Representative images of PNNs surrounding PV+ cells in the hippocampus and corresponding images of PNN units traced by the program in control and PLX-treated animals. The yellow frame indicates the chosen area shown with higher magnification on the right. Scale bar, 5 µm. (**B**) The factor analysis identifies the intensity and size of PNN units to be the most informative factors explaining 26% and 24% of the total variance, respectively. The scatter plot of individual PNN units in the space of these two factors reveals a fraction of PNN units with low-intensity ECM that disappears after PLX treatment. (**C**) Loads of measured variables on two principal factors. (**D**) Comparison of cumulative distribution functions to visualize an increase in ECM intensity and reduction in the hole area after PLX treatment. (**E**) Coefficient of variation analysis of ECM boundary perimeter shows that PNN units are more uniform per animal after PLX treatment: *** *p* = 0.001, *t*-test, n = 4 for both groups.

**Figure 4 cells-10-01862-f004:**
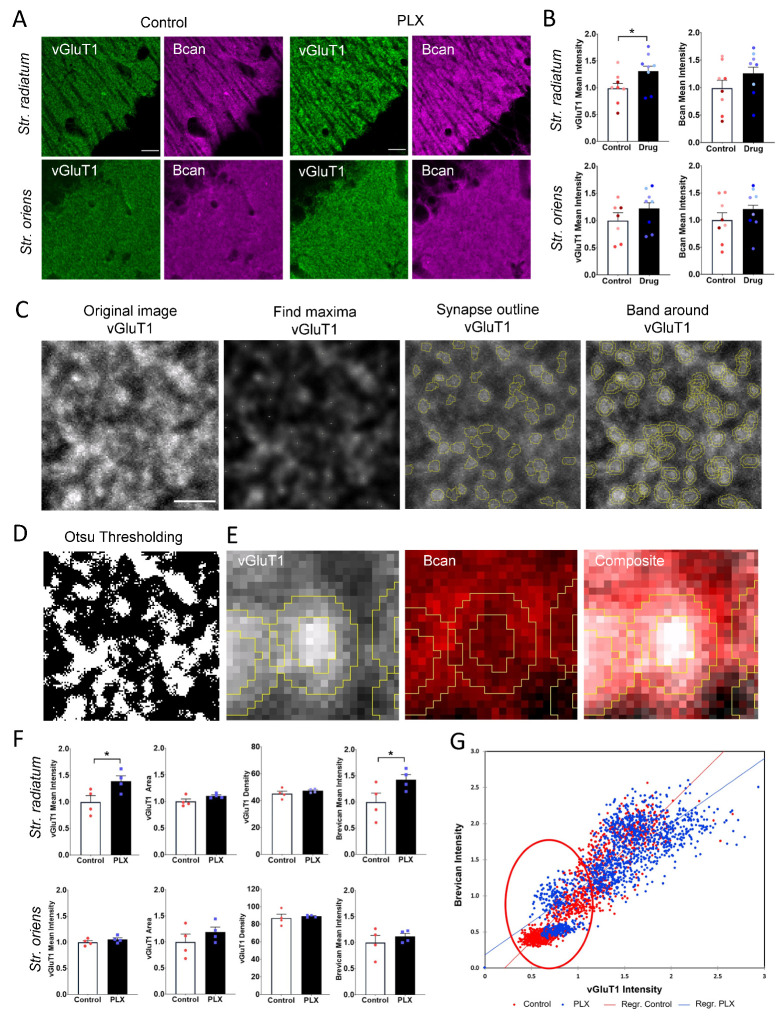
Elevation of presynaptic vGluT1 and perisynaptic Bcan after depletion of microglia. (**A**) Representative 63× images of adult hippocampal *str. radiatum* and *oriens* after PLX treatment and in control. (**B**) Quantification of vGluT1 and Bcan mean intensity per ROI shows upregulation in vGluT1 expression in *str. radiatum* (* *p* < 0.05, *t*-test). Bars show means per animal with superimposed individual values per ROI. Colors indicate ROIs from the same animal. (**C**) ROI showing the representative section of an image acquired with ×63 objective. (1) Original image after maximum projection. (2) After applying the Gaussian filter and enhancing contrast, the Find Maxima algorythm in Fiji ImageJ is used to detect local maxima. (3) Starting from the brightest pixel of each synapse, the outline of the puncta is detected. (4) Then, a band is created, and the mask is applied to the original image to measure size and intensity in the center and the band of the created ROI. (**D**) Using the conventional thresholding methods in Fiji, such as Otsu thresholding, we could not precisely detect synaptic puncta. (**E**) Zoomed image of single presynaptic punctum and surrounding ECM. (**F**) PLX3397 significantly increased the intensity of vGluT1 puncta and the intensity of surrounding perisynaptic brevican in *str. radiatum* (* *p* < 0.05, *t*-test), but not in *str. oriens*. The density of VGluT1 puncta was not changed. Bars and error bars represent mean ± SEM per animal. Single points on bars represent average values per animal. (**G**) The correlation plot shows a positive correlation between vGluT1 presynaptic intensity and corresponding perisynaptic brevican signal. Scale bar is 15 µm in (**A**) and 3 µm in (**C**–**E**).

**Figure 5 cells-10-01862-f005:**
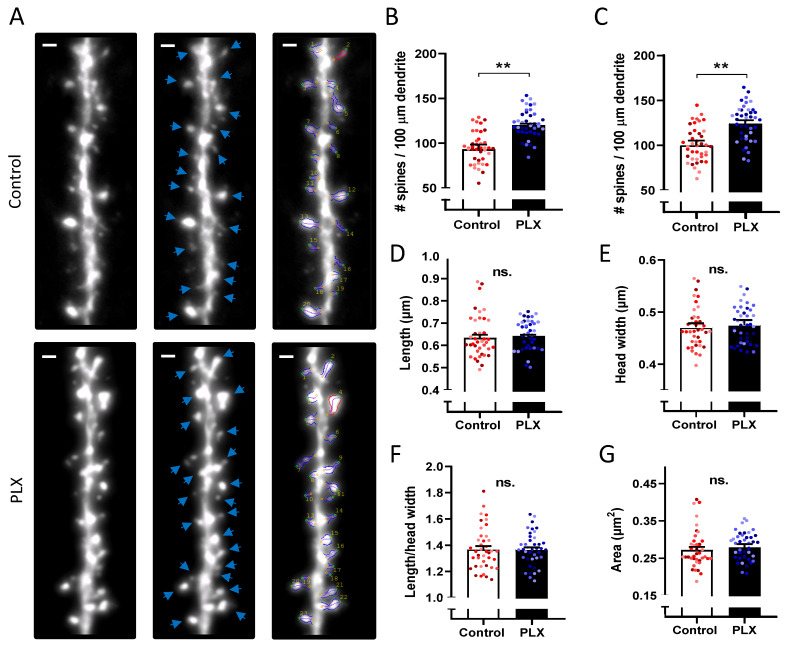
Increased spine density after depletion of microglia. (**A**) Representative 63× images of hippocampal secondary dendrites of control and PLX-treated animals in *str. radiatum*. Scale bar, 1 µm. (**B**–**G**) Quantification of spine morphology and density. Statistical analysis was performed using a nested t-test. Bars and error bars represent mean ± SEM values per animal. Single points on bars represent average values per cell, colors indicate cells from the same animal. After microglia depletion, the spine density is significantly higher as shown in (**B**) for the 2D analysis (** *p* = 0.0023) as well as in (**C**) for the 3D analysis (** *p* = 0.0087). (**D**–**G**) Quantification of spine morphology parameters revealed no differences between groups.

## Data Availability

Data supporting reported results can be obtained upon a request to the corresponding author (A.D.).

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
