# Peer review of "Microglia Depletion-Induced Remodeling of Extracellular Matrix and Excitatory Synapses in the Hippocampus of Adult Mice"

_cells, 2021, doi:10.3390/cells10081862_

Round 1

Reviewer 1 Report

In this study, authors examined the role of microglia in modulating PNNs, perisynaptic ECM, and excitatory synapses in hippocampus. They found that depletion of microglia by PLX3397 treatment altered the fine structure of PNN units without changing the number or intensity of PNNs. In addition, they found that depletion of microglia increased the intensity of presynaptic excitatory marker (vGlutI), and surrounding ECM (Bcan) and increased the density of dendritic spines.

Although microglia-mediated remodeling of synapses has been emerged, this study seemed to be incremental to the previous reports. A number of errors in the text alleviated the enthusiasm for the manuscript.

I have several comments on this manuscript as follows:

Major comments:

  1. All figures should be described in the Result section. (e.g. no description on Figure 2A)
  2. In Figure 3, A panel and B quantification graphs do not match.
  3. In Figure 3, the authors should present the vGluT1/Bcan images from the str. oriens.
  4. While Bcan intensity in str. orienswas not altered by PLX3397 (Figure 3), Bcan intensity appeared to be significantly increased (Figure 4). What makes this inconsistency? In Figure 4, the authors used more sophisticated synaptic puncta analyses than those used in Figure 3, claiming identical conclusion. I suggest that the authors have to apply the analyses used in Figure 4 to the data presented in Figure 3.
  5. The authors did not address how increase in vGluT1+ puncta intensity by PLX3397 is specifically observed in a certain layer. This should be mentioned and discussed.
  6. The authors investigated how elimination of microglia affects the intensity or number of the presynaptic excitatory markers and dendritic spine density, but this point is not specified in manuscript and no rationale is given. What about inhibitory synapses?
  7. The authors claimed that depletion of microglia reduced heterogeneity of PNN units with providing no direct evidence supporting it.

Minor comments:

  1. Throughout this manuscript, there are a number of typographical errors (e.g., PLX3397 is correct. PLX3377 or 3977 is wrong).
  2. Nomenclatures should be consistent throughout the manuscript.
  3. In Figure 1C, which color represents tdTomato? The authors should add the detailed information in the figure legend.
  4. Figure numbering in the Results section is incorrect throughout the manuscript (i.e., Figure 1A-C should be 1A-B in line 227; no panel F in Figure 1 (Line236))
  5. In Figure 1B, y axis dimension appears to be wrong (um2 should be replaced by mm2).
  6. In Figure 2A, the description on the four images with higher magnification on the right should be provided.

Author Response

Reviewer 1

In this study, authors examined the role of microglia in modulating PNNs, perisynaptic ECM, and excitatory synapses in hippocampus. They found that depletion of microglia by PLX3397 treatment altered the fine structure of PNN units without changing the number or intensity of PNNs. In addition, they found that depletion of microglia increased the intensity of presynaptic excitatory marker (vGlutI), and surrounding ECM (Bcan) and increased the density of dendritic spines.

Although microglia-mediated remodeling of synapses has been emerged, this study seemed to be incremental to the previous reports. A number of errors in the text alleviated the enthusiasm for the manuscript.

Reply: Our study is the first with the focus on quantitative analysis of perisynaptic ECM around perisomatic synapses on PV+ cells (PNN units) and around vGluT1+ excitatory synapses. There was some misunderstanding of our data, which we tried to correct. We hope that the restructuring of figures and other changes made in the revised manuscript will make the outcome of our study more clear and convincing and increase the enthusiasm of this reviewer.

I have several comments on this manuscript as follows:

Major comments:

  1. All figures should be described in the Result section. (e.g. no description on Figure 2A)

Reply: We thank the Reviewer for mentioning this and checked the manuscript. Now all the Figures are described in the results section.

  1. In Figure 3, A panel and B quantification graphs do not match.

Reply. We changed the image (that is now Fig. 2C) to better match the quantification shown. 

  1. In Figure 3, the authors should present the vGluT1/Bcan images from the str. oriens.

Reply: We added the images from the str. oriens as requested. This is now shown in Fig. 4A.

  1. While Bcan intensity in str. oriens was not altered by PLX3397 (Figure 3), Bcan intensity appeared to be significantly increased (Figure 4). What makes this inconsistency?

Reply: A large fraction of brevican is localized outside of perisynaptic ECM, so when total ROI analysis was used as in Fig. 3, we analyzed these two fractions together and the result may be dominated by non-perisynaptic ECM, which would explain the difference to the results of perisynaptic ECM analysis.

  1. In Figure 4, the authors used more sophisticated synaptic puncta analyses than those used in Figure 3, claiming identical conclusion. I suggest that the authors have to apply the analyses used in Figure 4 to the data presented in Figure 3.

Reply: The same images were used for the analyses shown in figs. 3 and 4.

  1. The authors did not address how increase in vGluT1+ puncta intensity by PLX3397 is specifically observed in a certain layer. This should be mentioned and discussed.

Reply: Our finding is not surprising, as there are fundamental differences between str. oriens as compared to str. radiatum. For instance, excitatory synapses on apical and basal dendrites in str. radiatum and oriens, respectively, differ significantly in extracellular and intracellular cascades that control synaptic gain: only synapses in the str. radiatum show a dependency on matrix metalloproteinase-9, which is known to be expressed in microglia (Brzdak et al., 2019). This is now stated in the Results.

The authors investigated how elimination of microglia affects the intensity or number of the presynaptic excitatory markers and dendritic spine density, but this point is not specified in manuscript and no rationale is given.

Reply: Microglia have been implicated in interactions with excitatory synapses in young adult brains and found to affect the formation and elimination of dendritic spines and presynaptic filopodia (Cangalaya et al., 2020 and refs herein). Hence, we analyzed the effects of microglia depletion in the dendritic spine density and the intensity or number of the presynaptic excitatory markers. This is now stated in the Results.

What about inhibitory synapses?

Reply: As ECM in PNNs is perisynaptic predominantly for GABAergic contacts, and PNN components tenascin-R and neurocan have been shown to modulate the number of such contacts (Nikonenko et al., 2003; Sullivan et al., 2018), the changes in PNN unit parameters are expected to affect inhibitory synapses. We added this point to the Discussion.

  1. The authors claimed that depletion of microglia reduced heterogeneity of PNN units with providing no direct evidence supporting it.

Reply: We originally showed these data in Fig. 2E. Now the data are shown in Fig. 3E and mentioned at the end of section 3.3: “…the analysis of the coefficient of variation revealed that PNN units after PLX treatment are more uniform regarding the size (Fig. 3E) (t-test, p = 0.001). “

Minor comments:

  1. Throughout this manuscript, there are a number of typographical errors (e.g., PLX3397 is correct. PLX3377 or 3977 is wrong).

Reply: Typographical errors were taken care of.

  1. Nomenclatures should be consistent throughout the manuscript.

Reply: It has been checked.

  1. In Figure 1C, which color represents tdTomato? The authors should add the detailed information in the figure legend.

Reply: We added the information about corresponding colors in the figure legend of that image, which is now a part of Fig. 2.

  1. Figure numbering in the Results section is incorrect throughout the manuscript (i.e., Figure 1A-C should be 1A-B in line 227; no panel F in Figure 1 (Line236))

Reply: We apologize for these mistakes and corrected the Figure numbering throughout the manuscript.

  1. In Figure 1B, y axis dimension appears to be wrong (um2 should be replaced by mm2).

Reply: We changed the dimension to mm2.

  1. In Figure 2A, the description on the four images with higher magnification on the right should be provided.

Reply: We were unsure what the Reviewer meant here and increased the size of panel legends, provided higher resolution inserts to show that the labeled cells have a morphology typical of microglia, and increased the size of panels with statistical comparison.

Reviewer 2 Report

The authors use drugs to deplete microglia in the mouse brain and then analyze the extracellular matrix in the hippocampus. Although the authors' research themes are interesting, there are major problems with their experimental methods, analysis methods, and subsequent considerations.

・ 4 animals in the control group and 4 animals in the drug administration group. It is too few to use only 4 animals in each group. 6 or more should be used.

・ Hippocampus is mainly classified into CA1, CA3, and DG. CA1 is further classified into so, sp, and sr. Each region of so, sp, and sr has completely different functions and structures, and it is necessary to classify them into so, sp, and sr for analysis in this experiment as well.

・ Basically, astrocytes have been considered as cells that secrete extracellular matrix in the brain, but this paper has no results or consideration regarding astrocytes.

・ There are no photographs or data showing that the drug depleted microglia in the brain.

・ PNN is composed of various components, and each component of the brain has different components. It is a big problem to always describe it as PNN. It is necessary to describe WFA + PNN etc. depending on the primary antibody used. 

Author Response

Reviewer 2

The authors use drugs to deplete microglia in the mouse brain and then analyze the extracellular matrix in the hippocampus. Although the authors' research themes are interesting, there are major problems with their experimental methods, analysis methods, and subsequent considerations.

・ 4 animals in the control group and 4 animals in the drug administration group. It is too few to use only 4 animals in each group. 6 or more should be used.

Reply: We understand the wish of reviewer to present the analysis of more animals. Unfortunately, this would take >9 months to receive a new ethical permission for these experiments (the previous one has expired), and to perform and analyze additional mice. Such a long delay in publication may affect its novelty and impact. Considering also a short time for resubmission given us by Editors, we presume that we shall proceed with the current data set. In the present study, we used advanced image analysis to increase the precision and reliability of our measurements. We make the scripts available to the neuro-community and this may stimulate the development of the field. It is possible that due to a limited number of animals used we missed some PLX effects, but the detected and reported effects are consistent and provide new information about the role of microglia in remodeling of perisynaptic ECM.

・ Hippocampus is mainly classified into CA1, CA3, and DG. CA1 is further classified into so, sp, and sr. Each region of so, sp, and sr has completely different functions and structures, and it is necessary to classify them into so, sp, and sr for analysis in this experiment as well.

Reply: Now we consistently performed the quantifications of excitatory presynaptic and perisomatic ECM markers in str. oriens and radiatum. CA1 str. pyramidale contains only few excitatory synapses, and these were not analyzed here.

・ Basically, astrocytes have been considered as cells that secrete extracellular matrix in the brain, but this paper has no results or consideration regarding astrocytes.

Reply: In a previous study, depletion of microglia was shown to induce no changes in GFAP+ cell numbers or morphology, but increased the expression levels of GFAP, suggesting an increased level of astrocyte activation (Elmore et al., 2014). Activated astrocytes secrete more CSPGs, which may then accumulate perisynaptically in line with the observed increase in Bcan after microglia depletion. We added this important information into Discussion.

・ There are no photographs or data showing that the drug depleted microglia in the brain.

Reply: These images have been presented in Fig. 1. To bring more clarity, we separated them from PNN images and show as Fig. 1A and B.

・ PNN is composed of various components, and each component of the brain has different components. It is a big problem to always describe it as PNN. It is necessary to describe WFA + PNN etc. depending on the primary antibody used. 

Reply: We agree and changed the labeling on the axis of Fig. 2D, in the Figure legend and in the text in Line 235 from PNN+ cells to WFA+ cells.

Reviewer 3 Report

By depleting microglia using CSF1R inhibitor PLX3397, the authors found that microglia regulate pre- and post-synaptic excitatory terminals and surrounding peri-synaptic extracellular matrix as well as the fine structure of perineuronal nets. This study is interesting. Some suggestions are listed as below:

  1. In line 57, ‘CNS’ should be ‘central nervous system (CNS)’. In line 62, ‘central nervous system’ should be ‘CNS’. Please double check throughout the manuscript.
  2. In lines 74-75, the authors said that CSF-1R is only expressed in microglia and not in neurons or other glial cells. This statement is not true. For example, previous studies suggested that effects on peripheral immunity should be considered both in interpretation of microglial depletion studies by systemic administration of CSF1R inhibition (PMID: 33203068 and PMID: 32900927). High-doses of PLX5622 and PLX3397 reduced oligodendrocyte progenitor cell number in primary cultures in vitro and ex vivo (PMID: 31029597). Furthermore, a small number of neurons may also express CSF1R (PMID: 27083478). Based on transcriptomics data some excitatory neurons can express the csf1r gene (the ALS Cell Atlas, http://alscellatlas.org/).
  3. The drug does of PLX3397 used in this study (1000 mg/kg for one month) is much higher than previous reports. Please discuss potential side-effects of the high dose of PLX3397 on other types of cells in the revised manuscript.
  4. Emerging data has convincingly demonstrated the existence of sex-dependent structural and functional differences of rodent microglia (PMID: 33731174). However, it is not clear for readers which gender of mice (male, female, or both) was used in this study? Did the authors observe the sex-dependent effect of microglial depletion on vGlut1 puncta and peri-synaptic Bcan in the CA1 region of the hippocampus?
  5. In Figure 1A and B, why the tdTomato+Iba1- cells (microglial?) were noted.
  6. I wonder if microglial repopulation following depletion can rescue the decreased size of PNN holes, increased the intensity of surrounding ECM, and reduced heterogeneity of PNN units.
  7. Did you observe any behavioral changes in these mice since CA1 is highly involved in learning and memory?
  8. In line 374, please correct the description 'While most studies so far were focused on the effects microglia have on PNNs'. Please double check throughout the manuscript.
  9. Potential mechanisms such as complement abnormalities should be discussed, for example, the C1q/C3-CR3 signaling-mediated synaptic remodeling.

Author Response

Reviewer 3

By depleting microglia using CSF1R inhibitor PLX3397, the authors found that microglia regulate pre- and post-synaptic excitatory terminals and surrounding peri-synaptic extracellular matrix as well as the fine structure of perineuronal nets. This study is interesting. Some suggestions are listed as below:

Reply: we greatly appreciate the interest of the reviewer in our data.

  1. In line 57, ‘CNS’ should be ‘central nervous system (CNS)’. In line 62, ‘central nervous system’ should be ‘CNS’. Please double check throughout the manuscript.

Reply: We introduced the requested changes.

  1. In lines 74-75, the authors said that CSF-1R is only expressed in microglia and not in neurons or other glial cells. This statement is not true. For example, previous studies suggested that effects on peripheral immunity should be considered both in interpretation of microglial depletion studies by systemic administration of CSF1R inhibition (PMID: 33203068 and PMID: 32900927). High-doses of PLX5622 and PLX3397 reduced oligodendrocyte progenitor cell number in primary cultures in vitro and ex vivo (PMID: 31029597). Furthermore, a small number of neurons may also express CSF1R (PMID: 27083478). Based on transcriptomics data some excitatory neurons can express the csf1r gene (the ALS Cell Atlas, http://alscellatlas.org/).

Reply: We greatly appreciate this comment and corrected the sentence in lines 74-75 and extended Discussion by mentioning possible effects of PLX on peripheral immunity,  oligodendrocyte progenitors and neurons.

  1. The drug dose of PLX3397 used in this study (1000 mg/kg for one month) is much higher than previous reports. Please discuss potential side-effects of the high dose of PLX3397 on other types of cells in the revised manuscript.

Reply: The daily dose reported here (1000 mg/kg of food with daily drug dose between 1-1.5 mg per animal) is similar with the dose used in combination with normal chow diet (290mg/kg of food but with daily drug dose between 0.8- 1.2 mg) (Elmore, 2014). In addition, no side-effects were reported using this dose in mouse models (Spiller et al. 2018; Cangalaya at al. 2020).

  1. Emerging data has convincingly demonstrated the existence of sex-dependent structural and functional differences of rodent microglia (PMID: 33731174). However, it is not clear for readers which gender of mice (male, female, or both) was used in this study? Did the authors observe the sex-dependent effect of microglial depletion on vGlut1 puncta and peri-synaptic Bcan in the CA1 region of the hippocampus?

Reply: The study was performed using male mice, we added this important information in Methods. In another unpublished study we observed effects of brevican on physiological  parameters in males rather than females, so this was the reason to use males.

  1. In Figure 1A and B, why the tdTomato+Iba1- cells (microglial?) were noted.

Reply: We used Cx3Cr1/CreTomato x GFPM Thy-1 mice. These mice express red fluorescent protein (tdTomato) in all microglia cells. To show that tdTomato indeed labels all microglia in the hippocampus, we performed confirmatory staining with the commonly used marker Iba1. In Fig. 1 we show tdTomato+, Iba1+ cells and their overlap.

  1. I wonder if microglial repopulation following depletion can rescue the decreased size of PNN holes, increased the intensity of surrounding ECM, and reduced heterogeneity of PNN units.

Reply: We agree this could be an interesting topic for a follow-up study.

  1. Did you observe any behavioral changes in these mice since CA1 is highly involved in learning and memory?

Reply: These were young adults, so no improvement in behavior was expected and these experiments were not performed.

  1. In line 374, please correct the description 'While most studies so far were focused on the effects microglia have on PNNs'. Please double check throughout the manuscript.

Reply: We rephrased that sentence: While previous studies focused on the effects microglia have on PNN components…

  1. Potential mechanisms such as complement abnormalities should be discussed, for example, the C1q/C3-CR3 signaling-mediated synaptic remodeling.

Reply: We appreciate the comment and added the following sentences into Discussion:

Increased spine density after microglia depletion suggests that these cells play a crucial role in the homeostatic maintenance of dendritic spine densities during adulthood (Rice et al., 2015). During development, microglia eliminate synapses via the complement pathway that mainly involve secreted C1q, C3 and C3Ra microglial receptor (Schafer et al., 2012). It is plausible that this pathway may also mediate microglia role in the maintenance of dendritic spines in adult animals.

Round 2

Reviewer 2 Report

The small number of mice is a very serious problem. It is not scientific that reproducibility is not ensured. 

Author Response

Reviewer 2

The small number of mice is a very serious problem. It is not scientific that reproducibility is not ensured.

Reply: The modest sample size might hinder detection of some of PLX effects. We now acknowledged this in Discussion: “Contrary to another study, we did not detect an overall increase in the numbers of PNNs after microglia depletion [23], which may be related to the modest sample size used in the present study. However, at the fine structure level, we observed robust alterations in the organization of PNN units.”

Most importantly from our point of view, the sample size used allowed us to detect several major effects of PLX on synapses and ECM with p<0.05. Using power calculations for the t-test with power=0.8, p=0.05 and the reported values of standard deviation (SD) for spine density (12% in Suzuki et al., 2020) and parameters characterizing the shape, size and intensity of perineuronal net units (<12%, Kaushik et al., 2020; Stoyanov et al., 2021), we estimated that with n=4 per group we would be able to detect >30% effects of PLX on these parameters. Also our previous simulations showed that differences in spine shape can be reliably detected with n=4 (Ruszczycki et al., 2012). Our previous analysis in the restrosplenial cortex with n=4 mice per group revealed multiple PLX effects on the parameters characterizing interactions between microglia and synapses (Cangalaya et al., 2020). Considering that the standard deviation of perisynaptic brevican expression measurements is about 20% (Mitloehner et al., 2020), sample size of 4 would allow us to reliably detect >50% differences in perisynaptic ECM with power=0.8 and p=0.05. We added this information to Methods, as also requested by Editors.

Reviewer 3 Report

The authors have addressed my concerns.

One minor point: The authors said that these mice express red fluorescent protein (tdTomato) in all microglial cells. However, in figure 1B, it seems that the number of tdTomato+ cells is a little bit higher than Iba1+ cells (at least, not equal). 

These experiments were performed at around P30 in male mice. I wonder if microglial depletion in adult mice may also cause an increase in presynaptic vGluT1 and perisynaptic ECM.

Author Response

The authors have addressed my concerns.

One minor point: The authors said that these mice express red fluorescent protein (tdTomato) in all microglial cells. However, in figure 1B, it seems that the number of tdTomato+ cells is a little bit higher than Iba1+ cells (at least, not equal).

Reply: The very minor difference in the total counts of tdTomato+ and Iba+ cells is within the counting error, which is unavoidable as some cell somata are “cut” in acquired optical sections and depending on intensity of labeling may fall below the detection limit.

These experiments were performed at around P30 in male mice. I wonder if microglial depletion in adult mice may also cause an increase in presynaptic vGluT1 and perisynaptic ECM.

Reply: As stated in the Methods, at P30 mice were injected intraperitoneally with tamoxifen for 5 consecutive days to induce the recombination that labels microglia with the red fluorescent protein tdTomato. The PLX3397 treatment to deplete microglia started when mice were adult (4 months of age). Thus, the reported in our study changes in the ECM and excitatory synapses are found in adult mice.